# Study on the Detection Method for Daylily Based on YOLOv5 under Complex Field Environments

**DOI:** 10.3390/plants12091769

**Published:** 2023-04-26

**Authors:** Hongwen Yan, Songrui Cai, Qiangsheng Li, Feng Tian, Sitong Kan, Meimeng Wang

**Affiliations:** College of Information Science and Engineering, Shanxi Agricultural University, Jinzhong 030801, China

**Keywords:** daylily, intelligent detection, complex environment in the field, YOLOv5, backbone network

## Abstract

Intelligent detection is vital for achieving the intelligent picking operation of daylily, but complex field environments pose challenges due to branch occlusion, overlapping plants, and uneven lighting. To address these challenges, this study selected an intelligent detection model based on YOLOv5s for daylily, the depth and width parameters of the YOLOv5s network were optimized, with Ghost, Transformer, and MobileNetv3 lightweight networks used to optimize the CSPDarknet backbone network of YOLOv5s, continuously improving the model’s performance. The experimental results show that the original YOLOv5s model increased mean average precision (mAP) by 49%, 44%, and 24.9% compared to YOLOv4, SSD, and Faster R-CNN models, optimizing the depth and width parameters of the network increased the mAP of the original YOLOv5s model by 7.7%, and the YOLOv5s model with Transformer as the backbone network increased the mAP by 0.2% and the inference speed by 69% compared to the model after network parameter optimization. The optimized YOLOv5s model provided precision, recall rate, mAP, and inference speed of 81.4%, 74.4%, 78.1%, and 93 frames per second (FPS), which can achieve accurate and fast detection of daylily in complex field environments. The research results can provide data and experimental references for developing intelligent picking equipment for daylily.

## 1. Introduction

Daylily is widely planted in both northern and southern China [1,2]. It can be used as daily food and has extensive medicinal value [3,4,5]. Therefore, the application of daylily in daily diet and medical research will attract more and more attention from researchers [6]. Picking is an important link in the development of the daylily industry. Timely picking can improve product quality and yield, as well as increase farmers’ income [7]. The existing picking robots are mainly common fruit and vegetable crops, such as apples, tomatoes, oranges, etc. Almost all of them have regular fruit shapes, which are convenient for picking with robotic arms. At present, no daylily picking robots were developed, so the picking of daylilies is conducted by hand finish. However, traditional manual picking is costly and inefficient, and because the flower buds of daylilies are relatively fragile if the manual picking method is improper, it will lead to problems such as damage to the flower buds and loss of nutrients. However, the intelligent picking of daylily cannot only reduce labor costs, reduce farmers’ labor, and improve farmers’ happiness in life, but also ensure product quality, promoting the intelligent development of the daylily industry. So intelligent picking of daylily is urgent. Object detection of daylily during the picking period is the key to achieving intelligent picking. In the complex field environment, problems such as branch occlusion, overlapping of multiple plants, and uneven lighting affect the efficiency of the intelligent detection of daylily and the accuracy and efficiency of picking [8]. Therefore, it is of great significance to research the detection methods of daylily in complex field environments to realize the intelligent picking of daylilies and promote the development of daylily picking robots [9].

The field of crop object detection achieved fruitful research results. In early studies, traditional methods such as support vector machines (SVM) [10,11], random forests [12], and HOG-SVM classifiers [13,14,15] were mainly used. Xiuxia Zhang et al. [16] designed a daylily robot recognition system based on ZYNQ. This system used a combination of traditional image processing and machine learning recognition algorithms to realize efficient real-time detection of daylilies; Jichao Zhao et al. [17] designed a daylily picking robot system, which uses a RGB-D binocular camera with depth function as the vision system. This system can effectively improve people’s labor production efficiency and avoid the harm caused by daylily picking on farmers’ health. Juanjuan Ma et al. [18] proposed an object detection method based on a deep-first random forest classifier; the method only splits one node in each recursive process, and the mAP of object detection on the SenseAndAvoid dataset can reach 69.3%. Kaibing Zhang et al. [19] proposed a method for diagnosing missing pixels in rapeseed leaves based on the HSV color histogram of segmented rapeseed leaf regions. The method extracted the HSV color histogram features of the segmented rapeseed leaf region and trained multiple support vector machine classifiers, achieving a mAP of 93% for missing pixel detection. Xin Guo et al. [20] proposed an apple multi-object detection method based on improved HOG and SVM, achieving a mAP of 90.46% for apple object recognition.

Studies showed that natural light and dark variations [21], climate changes [22], and the presence of overlapping plants, branch occlusion, and blurred images in complex field environments [23,24] can have a significant impact on the recognition efficiency of object detection methods. Traditional object detection methods often suffer from instability and low accuracy in complex field environments. With the widespread application of deep learning techniques [25,26,27,28] in agriculture, researchers used a series of object detection models, such as SSD [29], Faster R-CNN [30], and YOLO [31,32,33,34] to conduct crop detection research. Jingjing Tian [35] proposed an apple leaf disease detection method based on the SSD network, which improved the algorithm by establishing a multi-scale feature extractor, designing a V-space-assisted localization branch, and constructing a multi-scale attention mechanism module. The improved algorithm achieved a mAP of 83.42% for apple leaf disease detection. Gao et al. [36] proposed an apple detection method based on the Faster R-CNN network model using VGG-16 as the backbone network. The method achieved a mAP of 87.9% for apple detection. Ying Wang et al. [37] proposed an improved YOLOv5 algorithm for fruit and vegetable detection in complex environments, which improved the algorithm by embedding a convolutional block attention module (CBAM) in the CBL module of the backbone network Backbone, introducing a complete intersection over union non-maximum suppression (CIOU-NMS), and replacing the original YOLOv5 path aggregation network (PANet) with a weighted bidirectional feature pyramid network (BiFPN). The improved YOLOv5 algorithm achieved a mAP of 92.5%, which is 3.5% higher than the original YOLOv5 algorithm, enabling fast and accurate recognition of fruits and vegetables in complex environments. Zhu et al. [38] proposed an apple leaf disease detection model based on the improved YOLOv5 algorithm, which improved the algorithm by adding a feature enhancement module (FEM) and a coordinate attention (CA) method. The improved algorithm achieved a mAP of 95.9% for apple leaf disease detection. Wenxia Bao et al. [39] designed a wheat ear detection model based on YOLO with transformer prediction heads (TPH-YOLO), which improved the algorithm by adding the coordinate attention mechanism CA module in the backbone network of YOLOv5 and converting the original prediction head of YOLOv5 into Transformer prediction heads (TPH). The average precision of this method was 88.8%, which is 4.1% higher than that of the original YOLOv5. Zhouyi Xie et al. [40] proposed a multi-object flower recognition system based on YOLOv4, which replaced the original main feature extraction network with MobileNetv3 [41] and combined it with an optimized K-means clustering algorithm [42]. The improved YOLOv4 model achieved a precision rate of 96.43% for multi-object flower detection. Jie Liu et al. [43] proposed an orange recognition and positioning method based on the improved YOLOv4 model. The improved YOLOv4 model adopted MobileNetv2 as the backbone network and used depth-separable convolution in the neck structure instead of ordinary convolution. The optimized algorithm achieved a mAP of 97.24% for orange detection, reducing the average detection time by 11.39 ms and the model parameter amount by 197.5 M compared to the original YOLOv4 model. The results show that compared with traditional machine learning methods, such as SVM and random forest, deep learning techniques and convolutional neural networks have advantages in both speed and accuracy for crop object detection in agriculture.

According to the above analysis, although a series of achievements were made in crop detection based on deep learning, so far, there are few reports on the detection and identification of daylilies, and the existing detection models have limited applications in complex field environments. Therefore, this study proposes a detection method for daylily in complex field environments based on YOLOv5. In this study, 4200 naturally grown daylily samples were collected in different environments, such as sunny, cloudy, and nighttime according to the actual picking environment, the depth and width parameters of the YOLOv5 network were optimized, and the backbone network of the model was optimized to improve the model, which achieved high-precision real-time detection of daylilies at different growth stages in complex field environments. This method meets the requirements of real-time stability in actual picking scenarios and provides experimental reference and technical support for the development of intelligent picking of daylily.

## 2. Materials and Methods

### 2.1. Data Collection of Daylily

#### 2.1.1. Collection Equipment and Methods

There is no publicly available dataset for the identification of daylily during the picking period, so this experiment constructed its dataset of daylily. Daylily images were collected from late July to early August 2022 at three locations: the daylily plantation in Datong, Shanxi, the Daylily Park in Yunzhou District, Datong, Shanxi, and the Fenhe River in Sixian Village, Jinzhong, Shanxi. A handheld Canon EOS M100 camera was used to capture images of the flower buds of daylily (including the stems near the buds) at three angles: top view, flat view, and top view, and a total of 4200 images were captured. To consider the influence of natural light brightness and darkness in the field on the efficiency of the actual picking operation, the collected images included images of daylilies under sunny days, sunny backlights, cloudy days, and night light. At the same time, because the field environment is more complex, the collected images also include images under four conditions: overlapping plants, branch occlusion, blurred images, and uneven lighting. Finally, the images were saved in JPEG format, and the resolution of the images collected by this camera is 1920 × 1280 pixels, and the collected images are shown in Figure 1a–h.

#### 2.1.2. Data Preprocessing

The input image resolution of the YOLOv5 model used in this study is 640 × 640, while the image resolution of the daylily dataset is 1920 × 1280 pixels, and the length and width of the daylily target in the image are much smaller than the image size. Therefore, this study cuts the original image according to the position of the daylily flower bud center, sets the image size obtained by cutting the image to 640 × 640, and selects 3200 images from the cut daylily dataset as the original dataset. However, due to the complex field environment, too few datasets will lead to poor stability and generalization ability of the trained model. Studies showed that convolutional neural networks remain invariant to image transformations [44], and data augmentation can increase the number of images multiple times, enabling the model to achieve better training results, preventing over-fitting, and improving the generalization ability of the model. Therefore, this study augmented the original dataset and increased the number of pictures to 8000 through operations such as scaling and rotation, contrast adjustment, brightness adjustment, and mirror flip. The effect diagram of daylily before and after data augmentation is shown in Figure 2.

#### 2.1.3. Dataset Labeling

In this study, 4200 images were screened from the augmented dataset for labeling. The labeling tool was LabelImg, and the format of the label files was saved as XML files. According to the different growth stages of the daylily in the field environment, the daylily in the image is marked as Immature, Pluckable, Flowering, and Other classes, among which, the Immature and Pluckable classes are difficult to distinguish, and its characteristics are as follows: the Immature class is short in stature and a light green color; Pluckable class is characterized by full buds and golden color. The four types of samples in the daylily dataset are shown in Figure 3.

In this study, the 4200 labeled pictures were divided into the training set (2940 pictures), validation set (840 pictures), and test set (420 pictures) according to the ratio of 7:2:1. Among them, the number of pictures corresponding to Immature, Pluckable, Flowering, and Other classes are 3048, 1858, 1836, and 603, respectively. Since the images were taken in a natural environment, each image contains a variety of daylily classes; therefore, the total number of images of various classes is more than 4200. The distribution of four classes of labels in the daylily dataset is shown in Table 1.

### 2.2. YOLOv5 Network Model and Evaluation Indicators

Object detection algorithms based on deep learning can be divided into two categories: one-stage and two-stage. The two-stage object detection algorithm represented by Faster R-CNN needs to extract candidate boxes and classify candidate boxes to achieve object detection. It has the characteristic of high precision, but it requires a large amount of sample data and a long training time, and the detection speed is slow, so it is not suitable for scenes with high real-time requirements. The one-stage object detection algorithm represented by YOLO and SSD uses the convolutional layer and the input image to form the entire network structure. After the convolution operation, it directly returns the object category and position. It has the characteristic of real-time, but it is difficult to detect small targets and the accuracy is low. However, YOLOv5 is a representative algorithm model in the YOLO series. It has fast detection speed and good adaptability, and can automatically adapt to targets of different sizes. At the same time, it has excellent detection accuracy and strong scalability; it can be transformed on platforms such as GPU, CPU, and TensorFlow Lite. These characteristics make the YOLOv5 algorithm widely used in the field of object detection. Therefore, this study uses the YOLOv5 model for experiments. The comparison of one-stage and two-stage object detection algorithm frameworks is shown in Figure 4.

#### 2.2.1. YOLOv5 Model Principle and Structure

The network structure of YOLOv5 is divided into four parts, Input, Backbone, Neck, and Output. In the input part, the algorithm uses Mosaic [34] data augmentation, and performs rotation, scaling, and adaptive anchor frame processing on the input image [45] to increase the background information of the detection object and improve the algorithm’s performance in detecting small object performance. In the Backbone part, the YOLOv5 backbone network uses CSPDarknet53 [46]. The network structure is based on Darknet53 and uses the cross stage partial (CSP) module, which can effectively improve feature reuse and calculation efficiency. In the Neck part, the algorithm adopts the structure of the feature pyramid network (FPN) and path aggregation network (PAN) to enhance the feature aggregation capabilities of different feature layers, thereby improving the ability of the network to detect objects of different scales. In the Output part, YOLOv5 uses the GIOU_Loss loss function and weighted non-maximum suppression [47]. The network framework of YOLOv5 is shown in Figure 5.

The focus model and the spatial pyramid pooling (SPP) module are components of the Backbone. Among them, the focus module divides the input image into four smaller sub-images, then performs a convolution operation on each sub-image, and finally stitches the outputs of the four sub-images together to obtain the final output feature map. This approach not only reduces computation and memory consumption, but also improves model efficiency and accuracy. The SPP module is the last layer in Backbone. It adopts a spatial pyramid pooling structure and contains multiple pooling layers of different sizes, which can capture objects and features of different sizes. Without changing the size of the input image, the SPP module can improve. The receptive field and feature expression ability of the model can improve the detection performance of the model. The structure diagrams of the focus and SPP modules are shown in Figure 6 and Figure 7.

At the same time, the CSP module is also a component of the Backbone, and its structure consists of a convolutional layer and a residual block. The residual block consists of two convolutional layers and a cross-layer connection that adds the output of the first convolutional layer with the output of the second convolutional layer to form a residual block. The main idea of this model is to divide the input feature map into two parts, one of which performs convolution calculation, the other directly performs channel transformation, and finally merges the two parts. Using this method can reduce the repetition of information in the optimization process of the convolutional neural network and improve the learning performance of the network. The convolutional layer in the CSP module is composed of conv, batch normal, and SiLU activation functions, where the SiLU activation function formula is shown in Equation (1). The network hierarchy diagram of the CSP module is shown in Figure 8.
(1)$$silu\left(x\right)=x\times sigmoid\left(x\right)=\frac{x}{1+e^{-x}}$$

In YOLOv5, the Neck part adopts the FPN+PAN structure. Among them, the FPN structure can fuse feature maps of different levels to generate a feature pyramid with multi-scale information. The feature pyramid consists of three feature maps of different scales, which are: 1. 76 × 76 feature map—this is the bottom feature map with the highest resolution and can detect small-sized objects; 2. 38 × 38 feature map—this is the feature map of the second layer, the resolution is half lower than the 76 × 76 feature map, and it can detect objects of medium size; and 3. 19 × 19 feature map—this is the feature map of the highest layer, the resolution is the lowest, and it can detect objects of the largest size. The Neck part first upsamples the underlying feature map through the FPN module to obtain a series of feature maps of different scales. Then, the PAN module uses adaptive pooling and convolution operations to fuse feature maps of different scales to obtain a more complete and accurate feature representation. Finally, the object detector utilizes this feature representation for object detection and classification. By utilizing feature maps of different scales, the FPN+PAN structure can effectively improve the accuracy and robustness of object detection. The structure diagram of FPN+PAN is shown in Figure 9.

#### 2.2.2. Experimental Environment Settings

The operating system used in this experiment is Linux Ubuntu 18.04.5, using the Pytorch 1.9.0 framework for neural network training, the hardware configuration is 16 vCPU Intel(R) Xeon(R) Platinum 8350C CPU @ 2.60 GHz, the memory capacity is 56 GB, and the graphics card is NVIDIA GeForce RTX 3090, and the memory size is 24 GB.

#### 2.2.3. Model Parameter Settings and Evaluation Indicators

In this experiment, SSD, Faster R-CNN, YOLOv4, and YOLOv5 are used as the object detection model, and the SGD optimizer is used to optimize the network model. The image resolution of the input model is 640 × 640 pixels, the number of iterations set for model training is 100 times, the batch size for each iteration is set to 16, and the initial learning rate is set to 0.01. The specific model training hyperparameters are shown in Table 2.

The performance of object detection models is usually evaluated using indicators such as precision, recall, mean average precision (mAP), and frames per second (FPS). Among them, the precision rate can be used to evaluate the model’s ability to identify the detection object, and the recall rate can be used to evaluate the model’s ability to find positive sample objects. By recording the precision and recall values, a PR curve can be drawn. The average precision (AP) of the model detection object equals the area under the PR curve, and it can be used to evaluate the overall performance of the model for object detection and classification. The AP represents the average value of the average precision of all categories. Compared with the AP, the mAP can more accurately reflect the overall performance of the model in detecting various objects. The FPS refers to how many frames the network model can process per second, which is used to measure the speed at which the model processes images. Therefore, the evaluation of the object detection model in this experiment considers four indicators: precision, recall, mAP, and FPS. Among them, the calculation methods of *precision*, *recall*, *AP*, and *mAP* are shown in Equation (2) to Equation (5).
(2)$$Precision=\frac{TP}{TP+FP}$$
(3)$$Recall=\frac{TP}{TP+FN}$$
(4)$$AP=\int_{0}^{1}P(R)dr$$
(5)$$mAP=\frac{1}{n}\sum_{i=1}^{n}AP_{i}$$

Among them, *TP* represents the number of positive samples detected by the model, FP represents the number of negative samples detected by the model as positive samples, FN represents the number of positive samples detected by the model as negative samples, *P* is the precision rate, and *R* is the recall rate, while *P*(*R*) represents the maximum precision rate when the recall rate is r, n is the total number of classes (*n* = 4 in this experiment), and the value range of *i* is 1–6.

## 3. Experiment and Analysis

This experiment proposed a method based on the YOLOv5 model for daylily detection in complex field environments. First of all, the experiment trained and compared the mainstream object detection models, and selected the YOLOv5 model with the best effect. Then, the experiment optimized YOLOv5 by adjusting the depth and width of the model network and adjusted the Backbone of the YOLOv5 model to further improve the detection performance of the model. Finally, the experiment used the final optimized YOLOv5 model to detect the daylily in complex environments and performed a visual analysis of the detection results.

### 3.1. Basic Model Test of Daylily Detection

To test the performance of the YOLOv5 model in daylily detection, this experiment uses the daylily training set and test set divided in Section 2.1.3 of the article. First, the experiment uses the YOLOv5s, YOLOv4, SSD, and Faster R-CNN four original object detection models to train the training set. During training, the SGD optimizer is used, the learning rate is initialized to 0.01, the batch size is 16, and the number of training is 100 times. After the training is complete, the experiment uses the test set to test the trained model and calculate the precision, recall, mAP, and FPS of the four models. Last, the experiment compared their performance in daylily detection. The detection precision, recall, mAP, and FPS values of the four models for different growth stages of daylilies are shown in Figure 10.

It can be seen from Figure 10a that the precision of YOLOv5 and SSD models for the detection of immature, pickable, and flowering daylilies are all above 70%, while the precision of YOLOv4 for the detection of immature daylilies is slightly lower than 70%, and the detection accuracy of the Faster R-CNN model for daylilies is poor. It can be seen from Figure 10b–d that the YOLOv5s model shows obvious advantages in recall, mAP, and FPS compared with the other three models. The results of the specific evaluation indicators for the four models are shown in Table 3.

In Table 3, the precision of the YOLOv5s model is 72%, the recall is 67%, the mAP is 70.2%, and the inference speed is 178FPS. Compared with the YOLOv4, SSD, and Faster-RCNN models, the precision increased by 12.9, 9.9, and 36.8 percentage points, the recall increased by 60.2, 53.8, and 9.9 percentage points, the mAP increased by 49, 44, and 24.9 percentage points, and the inference speed increased by 2.7 times, 4.4 times, and 11.7 times. Experimental data show that the YOLOv5s model has higher detection precision and faster inference speed.

### 3.2. Parameter Optimization Experiment of Daylily Detection Model

According to the test results in the previous section, this study chose to use the YOLOv5s model to further improve the performance of the model in detecting daylily. Studies showed that the network depth and width coefficients of object detection models have an important impact on the performance and efficiency of the model. Mingxing Tan et al. [48] proposed a convolutional neural network structure EfficientNet with adjustable depth and width. The model can improve the performance and efficiency of the model by comprehensively adjusting the depth and width coefficients; Longxin Lin et al. [49] analyzed the impact of different scales of convolutional networks on the robustness and accuracy of object detection models through multiple experiments. The YOLOv5 model uses depth and width coefficients to control the size of the network. By increasing the depth and width coefficients, its feature extraction ability and object detection ability can be improved, and the effective receptive field of the model can be increased, which is beneficial for the model to better understand the visual information in the image and detect objects more accurately. Therefore, this experiment improves the network depth and width coefficients of the model. In this experiment, the configuration file of the model was named yolov5s.yaml, and then the network layer scaling factor (depth_multiple) and channel number scaling factor (width_multiple) of YOLOv5s were adjusted four times in the configuration file, and the adjusted models were named YOLOv5s1, YOLOv5s2, YOLOv5s3, and YOLOv5s4, respectively. The detailed parameters of the YOLOv5 (s, s1, s2, s3, and s4) model are shown in Table 4.

The network depth of YOLOv5s is 18 layers, and the number of channels is (32, 64, 128, 256, and 512). Therefore, the network depth of YOLOv5 (s1, s2, s3, and s4) models adjusted by the scaling factor can be calculated as 18 layers, 36 layers, 53 layers, and 72 layers; the numbers of channels are (16, 32, 64, 128, and 256), (48, 96, 192, 384, and 768), (64, 128, 256, 512, and 1024), (80, 160, 320, 640, and 1280), respectively. Subsequently, this experiment uses the adjusted models to train on the training set according to the same training parameters in Section 3.1 and selects the best weight trained by each adjusted model for performance testing on the test set. Finally, they are compared with the original YOLOv5s model. The precision, recall, mAP, and FPS results are shown in Table 5.

In Table 5, the precision of the YOLOv5 (s1, s, s2, s3, and s4) models for daylily detection are 70.3%, 72%, 80.8%, 82.9%, and 83%, for recall are 62.9%, 67%, 71.2%, 72.1%, and 72.8%, for mAP are 67.5%, 70.2%, 75.8%, 77.7%, and 77.9%, and the inference speeds are 213FPS, 178FPS, 169FPS, 167FPS, 55FPS. Compared with YOLOv5s, the mAP of YOLOv5s1 is reduced by 2.7 percentage points, but the inference speed is 20 percentage points faster; the mAP of YOLOv5s2 is increased by 5.6 percentage points, but the inference speed is 5 percentage points slower; the mAP of YOLOv5s3 is increased by 7.5 percentage points, but the inference speed is 7 percentage points slower; and the mAP of YOLOv5s4 is increased by 7.7 percentage points, but the inference speed is 2.2 times slower, with the highest detection precision and the slowest inference speed.

To better show the changing trend of model performance, this experiment compares the trained results visually. The precision, recall, mAP, and FPS comparison of the five models are shown in Figure 11.

In Figure 11, at the initial stage of training, as the number of iterations increases, the precision, recall, and mAP values of the five models all show a rapid increase trend. When the number of iterations reaches 73, the performance indicator values of the five models gradually tend to be stable. Among them, YOLOv5s4, which has the largest network structure, exhibits the fastest performance improvement and the highest mAP value. The comparison of the model detection performance of YOLOv4, SSD, Faster R-CNN, and YOLOv5 (s1, s, s2, s3, and s4) is shown in Table 6.

It can be seen from Table 6 that compared with YOLOv5s, the mAP of YOLOv4, SSD, Faster R-CNN, and YOLOv5s1 decreased by 49, 44, 24.9, and 2.7 percentage points, respectively, and the mAP of YOLOv5s2, YOLOv5s3, and YOLOv5s4 increased by 5.6, 7.5, and 7.7, respectively. It is worth noting that for the YOLOv5 (s1, s, s2, s3, and s4) model, as the model network depth and width coefficients increase, the precision, recall, and mAP of the model are continuously improved, but the magnitude of the model performance improvement gradually decreases, and the inference speed gradually slows down. Therefore, this study chose YOLOv5s4 for subsequent optimization experiments.

### 3.3. Backbone Network Optimization Experiment of Daylily Detection Model

In the optimization experiment of network parameters, the YOLOv5s4 model has the highest detection precision, and its mAP can reach 77.9%, but its inference speed is low, only reaching 55FPS. In the application of actual detection models, the lightweight optimization of the backbone network can improve the detection performance and inference speed of the model. Ghost, Transformer, and MobileNetv3 are currently three common lightweight networks. Among them, Transformer and Ghost networks have higher accuracy, but their speed is slower than MobileNetv3; the MobileNetv3 network speed is faster, but its accuracy is not high. Therefore, it is necessary to select the corresponding network for optimization according to the requirements of specific scenarios, such as accuracy, speed, and resource constraints. Lei Huang et al. [50] proposed a new FS-MobileNetV3 network to replace the CSPDarknet backbone network in the original network. Compared with the original model, the mAP of the improved model is only reduced by 0.37 percentage points, but the detection speed is improved by 11FPS, which meets the needs of mobile deployment in different scenarios; Xiaoqiang Shao et al. [51] used the improved lightweight network ShuffleNetV2 to replace the original YOLOv5s backbone network CSPDarknet53 and integrated the Transformer self-attention module into the improved ShuffleNetV2. Compared with the original YOLOv5s model, the mAP of the improved detection model increased by 5.2 percentage points, and the speed increased by 21 percentage points. Based on these research results and the optimization experiment results in the previous section, this study named the configuration file of the optimized model as yolov5s4.yaml, and replaced the backbone network CSPDarknet of the YOLOv5s4 model with lighter backbone networks Ghost [52], Transformer [53], and MobileNetv3 in the configuration file. Finally, they are trained and tested according to the same parameters in Section 3.1. The precision, recall, mAP, and FPS indicator results of daylily detection by different backbone networks of YOLOv5s4 are shown in Table 7.

In Table 7, compared with the YOLOv5s4 model with CSPDarknet as the backbone network, the mAP of the YOLOv5s4 model based on MobileNetv3 is reduced by 19.3 percentage points, but the inference speed is 1.9 times faster; the mAP of the YOLOv5s4 model based on Ghost is reduced by 8.3 percentage points, but the inference speed is 56.4 percentage points faster. While the mAP of the YOLOv5s4 model based on Transformer is increased by 0.2 percentage points, and the inference speed is 69 percentage points faster, the recall, mAP, and inference speed of the model are improved, and the detection performance is the best. Therefore, this experiment chooses the YOLOv5s4 model with Transformer as the backbone network as the final optimization model. The optimization results of the Transformer backbone network for the YOLOv5s4 model are shown in Table 8.

It can be seen from Table 8 that the detection performance of the YOLOv5s4 model with Transformer as the backbone network is reduced by only 1.5% in precision, and the recall, mAP, and inference speed are increased by 1.6%, 0.2%, and 38FPS, respectively, and the backbone network optimization experiment achieved expectations. To better demonstrate the changing trend of model performance based on different backbone networks, the trained results were visualized and compared. The precision, recall, mAP, and FPS of YOLOv5s4 model detection based on different backbone networks are shown in Figure 12.

### 3.4. Visual Analysis of the Detection Results of the Final Optimization Model on the Test Set 

To investigate the algorithm robustness of the YOLOv5 model for detecting daylily in complex field environments, this experiment selected images of daylily with uneven lighting, overlapping plants, night lights, and branch occlusion in the test set. The final optimized model was used for object detection, and the detection results are shown in Figure 13.

In Figure 13, the improved YOLOv5 model has a good detection effect in complex field environments. In the case of uneven lighting, the detection precision of the model for the Immature class can reach up to 96%, and for the Pluckable class can reach up to 97%. In the case of overlapping plants, the detection precision for the Immature class can reach up to 93%, and for the Pluckable class can reach up to 97%. In the case of night light, the detection precision for the Immature class can reach up to 94%, for the Pluckable class can reach 94%, and for the Flowering class can reach 95%. In the case of branch occlusion, the detection precision for the Immature class can reach up to 92%, and for the Pluckable class can reach up to 98%.

## 4. Discussion

### 4.1. Analysis of the Difference in Detection Performance of the Final Model for Daylily in Different Growth Stages

The improved YOLOv5 object detection model proposed in this study achieves real-time and efficient identification of daylilies at different growth stages in complex field environments. The detection effect for the Immature class is the best, with the mAP of 85.8%. The mAP for the Pluckable and Flowering classes are 85.5% and 84.8%, respectively, and the inference speed can reach 93FPS, meeting the precision and speed requirements for actual picking. However, the individual differences in the growth of daylilies in the natural environment will cause the daylilies in the same planting period to be in different growth stages, which may affect the detection accuracy of the model. Therefore, future research can consider increasing the data of daylilies in different growth stages to train the model to ensure the accuracy and diversity of the data, and then improve the model’s ability to detect daylilies in different growth stages. This will improve the efficiency of intelligent picking and sorting of daylilies, thereby improving the production efficiency of the daylily industry.

### 4.2. Analysis of the Impact of Different Optimization Methods on Object Detection Performance

At present, the demand for embedded products and services is increasing day by day. For embedded systems with limited memory space, object detection models not only require high-precision recognition effects, but also require the smallest possible network scale. This study adopted two methods to optimize the YOLOv5 model. First, when optimizing the network parameters of the YOLOv5 model, the network size of YOLOv5s is smaller than that of YOLOv5s4. Although its detection accuracy is lower than that of the YOLOv5s4 model, its reasoning speed is faster, which is conducive to deployment on mobile devices and other lightweight devices; while the YOLOv5s4 model has excellent detection accuracy, its network model is too large to be embedded in hardware devices. Second, when optimizing the backbone network of the YOLOv5 model, replacing the lightweight backbone network can effectively reduce the size of the model parameters, but the recognition accuracy of the model is also reduced. Although the YOLOv5s4 model based on Transformer improved significantly in speed without decreasing the recognition accuracy, there is still a big gap with the inference speed of the YOLOv5s model. Studies showed that adding an attention mechanism can effectively improve the detection accuracy of the model [54,55], and pruning the model redundant network can effectively compress the model network size [56]. Therefore, in future research, the following measures can be taken to further improve the YOLOv5 model: on the one hand, the detection accuracy of the YOLOv5s model can be improved by adding an attention mechanism. On the other hand, redundant neurons and weights in the neural network of the final optimized model can be pruned to improve the algorithm reasoning speed.

## 5. Conclusions

This study aims to achieve efficient detection of daylilies at different growth stages in complex field environments. Four mainstream object detection algorithms are used for training and comparison, and the YOLOv5s model with the best detection performance is selected for experimentation. On this basis, this study made the following improvements to the YOLOv5s model: firstly, adjust the depth and width coefficients of the YOLOv5s network, and then replace the YOLOv5s backbone network CSPDarknet with Ghost, Transformer, and MobileNetv3 lightweight networks, respectively, which realize high-precision real-time detection of daylilies in complex field environments. The conclusions are as below:(1)The mAP of YOLOv5s can reach 70.2%, and the inference speed can reach 178FPS. Compared with SSD, Faster R-CNN, and YOLOv4 models, it has higher detection accuracy and inference speed;(2)Adjusting the depth and width coefficients of the YOLOv5s network and optimizing the backbone network can further improve detection precision and inference speed. Among them, the YOLOv5s model with a Transformer-based network depth of 1.33 and a width of 1.25 has the best detection performance, and its mAP is 78.1%. The inference speed is 93FPS, and compared with the original YOLOv5s model, the mAP increased by 7.9 percentage points; compared with the YOLOv5s model based on CSPDarknet with the same network parameters, the mAP increased by 0.2 percentage points, and the inference speed increased by 69 percentage points;(3)Under the influence of occlusion, overlapping, visual blur, natural light bright or dark, weather, and other factors, the final optimized YOLOv5 model can still efficiently detect daylilies, and the mAP of the Immature, Pluckable, and Flowering classes, which, respectively, reached 85.8%, 85.5%, and 84.8%; this method has good stability and can meet the requirements of daylily picking operations. This study can provide a certain technical reference for the detection of crops in similar environments and the development of intelligent picking of daylilies.

## Figures and Tables

**Figure 1 plants-12-01769-f001:**
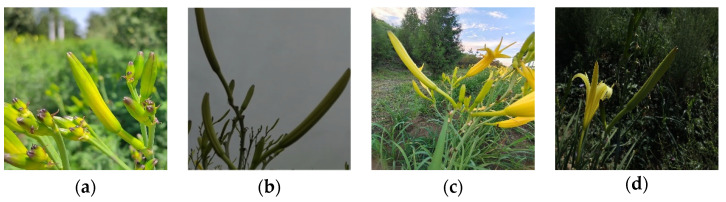
Images of daylily collected in the complex field environment. (**a**) Sunny day; (**b**) cloudy day; (**c**) sunny backlight; (**d**) night light; (**e**) overlapping plants; (**f**) branch occlusion; (**g**) blurred image; and (**h**) uneven lighting.

**Figure 2 plants-12-01769-f002:**
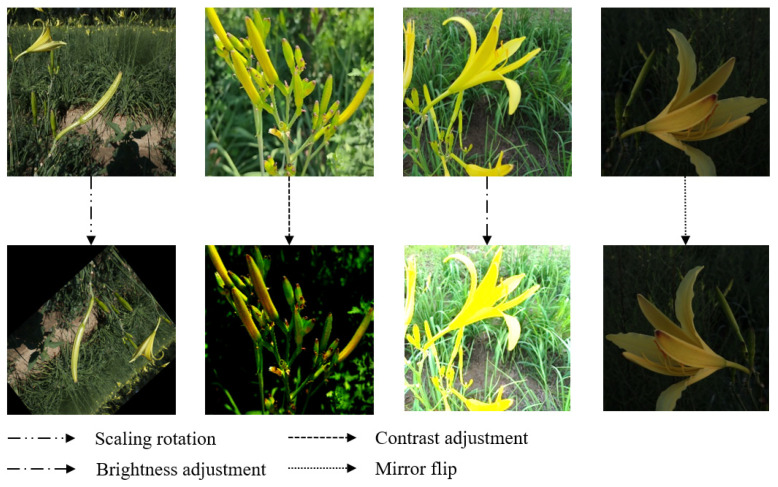
Data augmentation on daylily images.

**Figure 3 plants-12-01769-f003:**
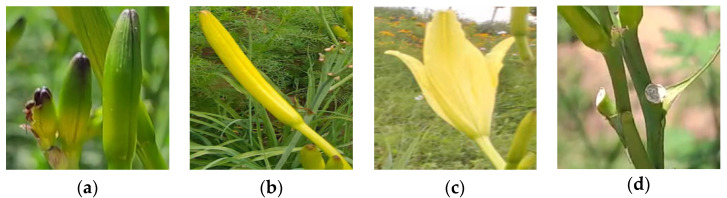
Display of four classes of samples in the daylily dataset. (**a**) Immature; (**b**) Pluckable; (**c**) Flowering; and (**d**) Other.

**Figure 4 plants-12-01769-f004:**
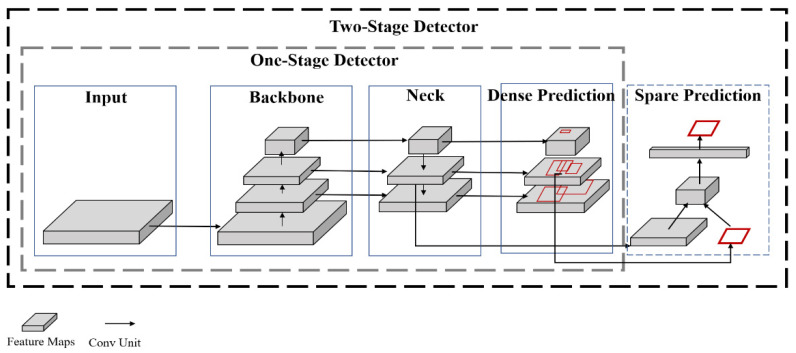
Comparison chart of one-stage and two-stage object detection algorithm frameworks.

**Figure 5 plants-12-01769-f005:**
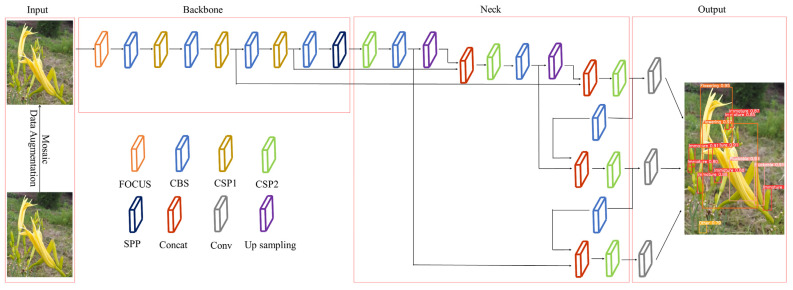
YOLOv5 network framework.

**Figure 6 plants-12-01769-f006:**
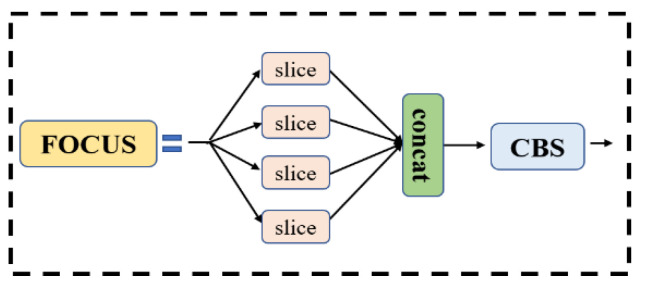
Focus structure.

**Figure 7 plants-12-01769-f007:**
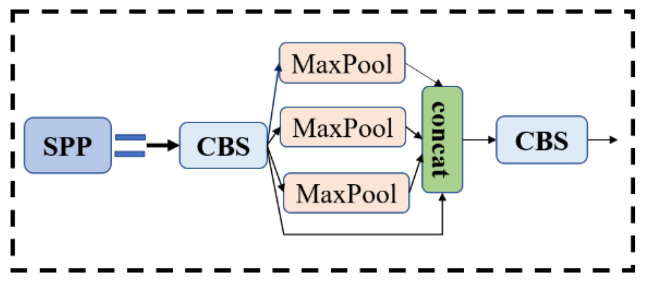
SPP structure.

**Figure 8 plants-12-01769-f008:**
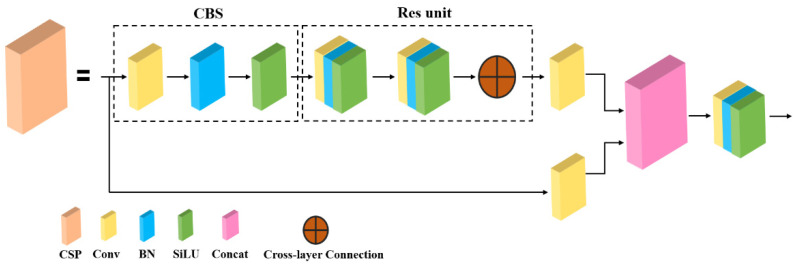
CSP model network hierarchy diagram.

**Figure 9 plants-12-01769-f009:**
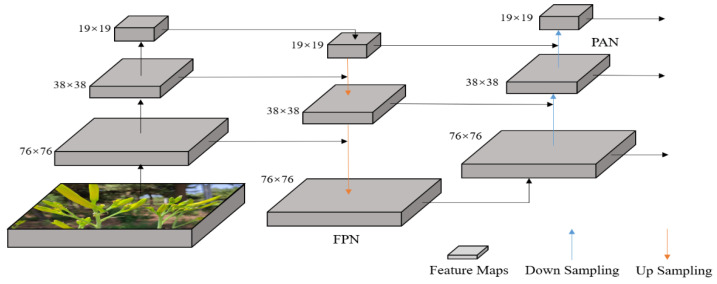
FPN+PAN structure diagram.

**Figure 10 plants-12-01769-f010:**
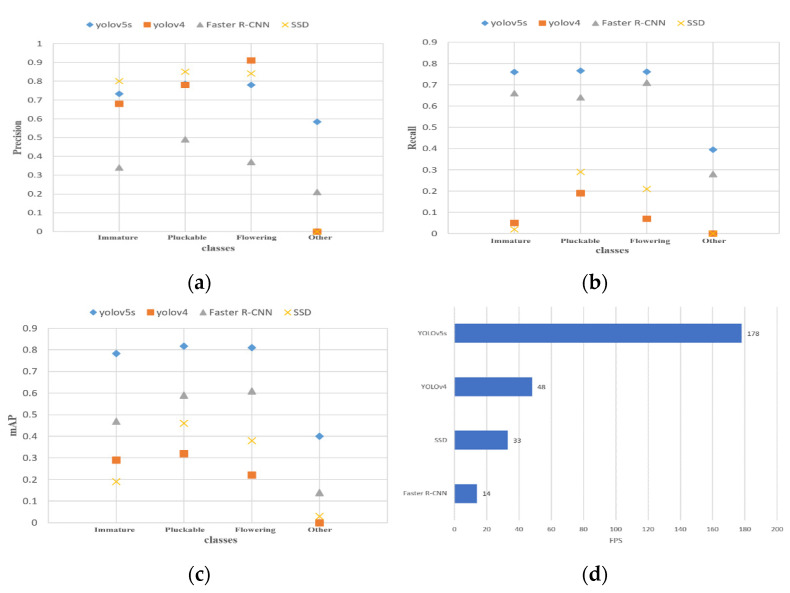
Comparison of YOLOv5s, YOLOv4, SSD, Faster R-CNN model detection results. (**a**) Precision; (**b**) recall; (**c**) mAP; and (**d**) FPS.

**Figure 11 plants-12-01769-f011:**
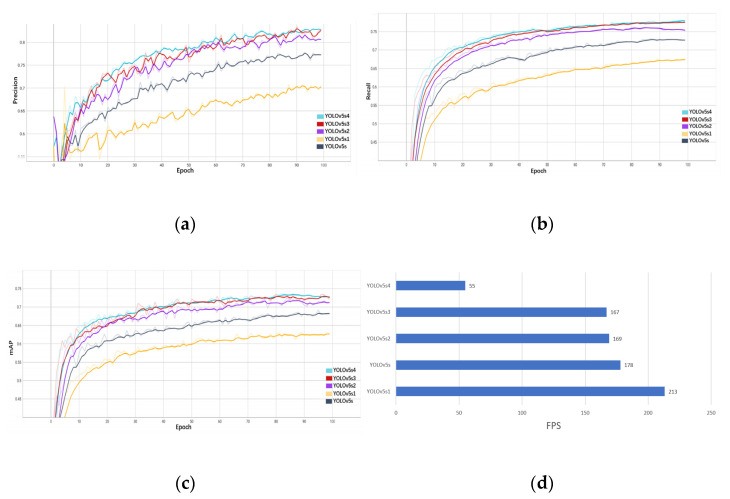
Comparison results of YOLOv5 (s, s1, s2, s3, and s4) model detection performance. (**a**) Precision; (**b**) recall; (**c**) mAP; and (**d**) FPS.

**Figure 12 plants-12-01769-f012:**
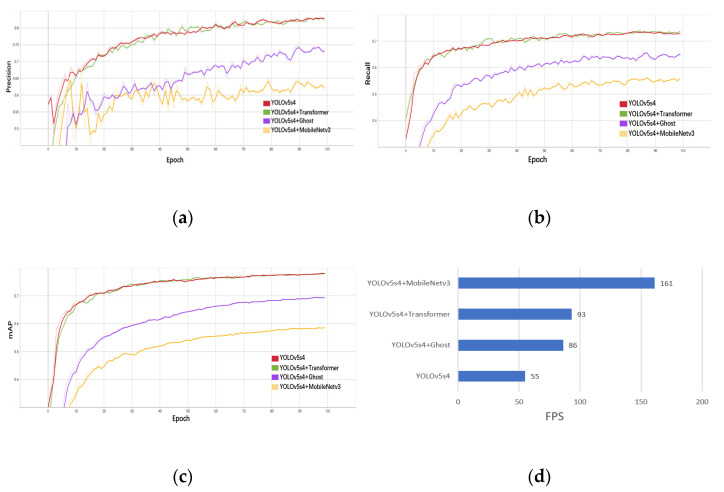
Comparison results of YOLOv5s4 model detection performance based on different backbone networks. (**a**) Precision; (**b**) recall; (**c**) mAP; and (**d**) FPS.

**Figure 13 plants-12-01769-f013:**
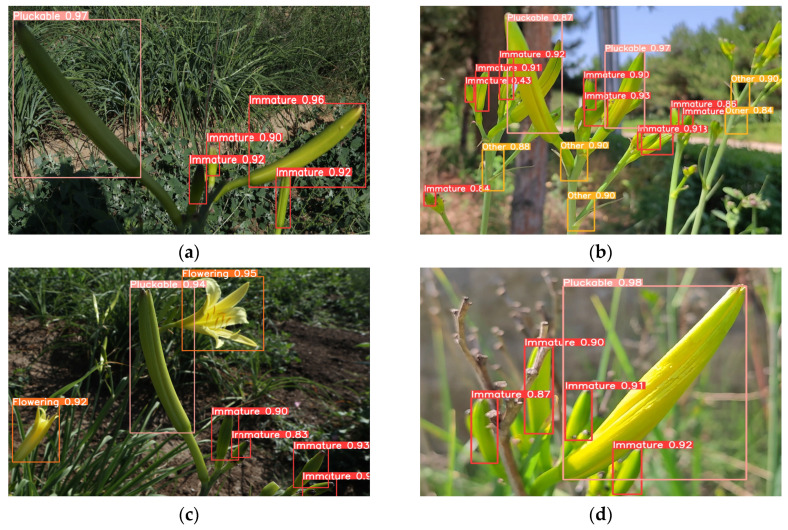
Visualization of object detection results. (**a**) Uneven lighting; (**b**) overlapping plants; (**c**) night lights; and (**d**) branch occlusion.

**Table 1 plants-12-01769-t001:** Distribution table of four classes of labels in the daylily dataset.

Classes	Training Set Labels	Validation Set Labels	Test Set Labels	Training Set	Validation Set	Test Set
Immature	31,551	8048	4504	2134	610	304
Pluckable	8901	2249	1270	1301	372	185
Flowering	8245	2081	1176	1285	368	183
Other	5401	1369	768	422	121	60
Total	54,098	13,747	7718	2940	840	420

**Table 2 plants-12-01769-t002:** Model training parameters.

Parameters	Value
Input image resolution	640 × 640
Iterations	100
Batch size	16
Initial learning rate	0.01
Learning rate momentum	0.937
Weight decay coefficient	5 × 10^−4^
IoU threshold	0.2

**Table 3 plants-12-01769-t003:** Results of YOLOv5s, YOLOv4, SSD, and Faster R-CNN model tests.

Model	Classes	Precision (%)	Recall (%)	mAP (%)	FPS
YOLOv5s	Immature	73.2	76.0	78.2	178
	Pluckable	78.5	76.6	81.7	
	Flowering	78.0	76.1	81.0	
	Other	58.3	39.5	40.0	
	All	72.0	67.0	70.2	
YOLOv4	Immature	68.0	4.6	29.0	48
	Pluckable	78.1	18.7	32.0	
	Flowering	90.7	6.8	22.0	
	Other	0	0	0	
	All	59.1	6.8	21.2	
SSD	Immature	80.0	2.2	19.0	33
	Pluckable	85.1	29.0	46.0	
	Flowering	83.7	21.0	38.0	
	Other	0	0	3.0	
	All	62.1	13.2	26.2	
Faster R-CNN	Immature	33.9	65.8	47.0	14
	Pluckable	49.3	64.2	59.0	
	Flowering	36.9	71.5	61.0	
	Other	20.6	28.0	14.0	
	All	35.2	57.1	45.3	

**Table 4 plants-12-01769-t004:** YOLOv5 (s, s1, s2, s3, and s4) model network parameters.

Model	Depth_Multiple	Width_Multiple	Params (M)
YOLOv5s	0.33	0.5	7.2
YOLOv5s1	0.33	0.25	1.9
YOLOv5s2	0.67	0.75	21.2
YOLOv5s3	1.0	1.0	46.5
YOLOv5s4	1.33	1.25	86.7

**Table 5 plants-12-01769-t005:** YOLOv5 (s, s1, s2, s3, and s4) model detection performance indicators results.

Model	Classes	Precision (%)	Recall (%)	mAP (%)	FPS
YOLOv5s	Immature	73.2	76.0	78.2	178
	Pluckable	78.5	76.6	81.7	
	Flowering	78.0	76.1	81.0	
	Other	58.3	39.5	40.0	
	All	72.0	67.0	70.2	
YOLOv5s1	Immature	73.4	73.2	77.0	213
	Pluckable	76.2	73.7	80.1	
	Flowering	76.7	72.3	78.5	
	Other	54.9	32.2	34.4	
	All	70.3	62.9	67.5	
YOLOv5s2	Immature	81.7	79.4	83.5	169
	Pluckable	83.3	78.5	84.5	
	Flowering	85.3	78.9	83.5	
	Other	72.9	47.9	51.9	
	All	80.8	71.2	75.8	
YOLOv5s3	Immature	84.7	80.1	84.3	167
	Pluckable	84.0	79.7	85.6	
	Flowering	87.4	79.0	85.1	
	Other	75.5	49.8	55.7	
	All	82.9	72.1	77.7	
YOLOv5s4	Immature	84.4	80.8	85.4	55
	Pluckable	82.7	82.0	85.4	
	Flowering	85.8	78.6	84.1	
	Other	78.6	49.7	56.7	
	All	83.0	72.8	77.9	

**Table 6 plants-12-01769-t006:** Performance comparison of different object detection models.

Model	YOLOv4	SSD	Faster R-CNN	YOLOv5s	YOLOv5s1	YOLOv5s2	YOLOv5s3	YOLOv5s4
Precision (%)	59.1	62.1	35.2	72.0	70.3	80.8	82.9	83.0
Precision Change	−12.9	−9.9	−36.8	0	−1.7	+8.8	+10.9	+11.0
Recall (%)	6.8	13.2	57.1	67.0	62.9	71.2	72.1	72.8
Recall Change	−60.2	−53.8	−9.9	0	−4.1	+4.2	+5.1	+5.8
mAP (%)	21.2	26.2	45.3	70.2	67.5	75.8	77.7	77.9
mAP Change	−49.0	−44.0	−24.9	0	−2.7	+5.6	+7.5	+7.7
FPS	48	33	14	178	213	169	167	55
FPS Change	−130	−145	−164	0	+35	−9	−11	−123

**Table 7 plants-12-01769-t007:** YOLOv5s4 different backbone networks for daylily detection performance indicator results.

Model	Backbone	Classes	Precision (%)	Recall (%)	mAP (%)	FPS
YOLOv5s4	CSPDarknet	Immature	84.4	80.8	85.4	55
		Pluckable	82.7	82.0	85.4	
		Flowering	85.8	78.6	84.1	
		Other	78.6	49.7	56.7	
		All	82.9	72.8	77.9	
YOLOv5s4	Ghost	Immature	76.6	73.1	78.2	86
		Pluckable	78.6	74.3	81.1	
		Flowering	80.7	74.5	81.6	
		Other	59.7	35.4	37.4	
		All	73.9	64.3	69.6	
YOLOv5s4	Transformer	Immature	82.8	82.9	85.8	93
		Pluckable	80.9	82.3	85.5	
		Flowering	85.0	81.3	84.8	
		Other	76.9	51.2	56.2	
		All	81.4	74.4	78.1	
YOLOv5s4	MobileNetv3	Immature	64.8	67.7	68.8	161
		Pluckable	71.1	68.6	74.1	
		Flowering	68.0	71.0	74.0	
		Other	46.4	15.5	17.5	
		All	62.6	57.7	58.6	

**Table 8 plants-12-01769-t008:** Performance comparison of different object detection models.

Model	Precision (%)	Precision Change	Recall (%)	Recall Change	mAP (%)	mAP Change	FPS	FPS Change
YOLOv5s4	82.9	0	72.8	0	77.9	0	55	0
Transformer + YOLOv5s4	81.4	−1.5	74.4	+1.6	78.1	+0.2	93	+38

## Data Availability

Not applicable.

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
