# Peer review of "Study on the Detection Method for Daylily Based on YOLOv5 under Complex Field Environments"

_plants, 2023, doi:10.3390/plants12091769_

Round 1

Reviewer 1 Report

The manuscript presents Daylily detection and classification study with popular YOLOv5 deep learning model.
The authors report the model architecture thoroughly and include demonstrative figures. Also the dataset and results are presented comprehensively. Experiments are performed with different models, and also with different depths and backbones for YOLOv5 model. The article shows a well justified object detection solution for improving intelligent picking of Daylilies, and can help in development of similar solutions for other crops, too.

Minor fixes:
-Figure 5 (line 187) is blurry and text is difficult to read. Especially the names of different layers should be clearer.
-In Figure 10 (line 283), the graphs (a)-(c) can be improved. As one line represents single scores for each of the 4 classes, the graphs would be more legible with markers instead of lines. Also, the classes should be in same order in the graphs as they are in all the tables (Immature, Pluckable, Flowering, Other) for uniformity.

Reviewer 2 Report

Overall, the paper presents an interesting approach for improving intelligent detection of Daylily using the YOLOv5s model with optimized depth and width parameters and the use of lightweight networks to optimize the CSPDarknet backbone network. The experimental results show significant improvements in Mean Average Precision (mAP) compared to other models and demonstrate the potential of the optimized YOLOv5s model for accurate and fast detection of Daylily in complex field environments.

However, there are several major revision comments to consider:

  1. The paper would benefit from a clearer motivation for the research. Why is it important to achieve accurate and fast detection of Daylily in complex field environments? What are the potential applications of this research?

  2. The introduction section needs to provide more background information on the current state of research in intelligent detection of Daylily and the challenges posed by complex field environments. The authors should also discuss the limitations of existing models and why the YOLOv5s model was chosen for this study.

  3. The methodology section should be more detailed and provide step-by-step instructions for replicating the experiments. The authors should also explain why they chose to optimize the depth and width parameters of the YOLOv5s model and how they selected the Ghost, Transformer, and MobileNetv3 lightweight networks to optimize the CSPDarknet backbone network.

  4. The results section needs to be reorganized and presented in a clearer format. The authors should provide a table summarizing the performance of each model tested (YOLOv5s, YOLOv4, SSD, and Faster R-CNN) and the improvements in mAP achieved by the optimized YOLOv5s model. The authors should also provide a table summarizing the performance of the optimized YOLOv5s model with and without the use of the Transformer backbone network.

  5. The authors should discuss the limitations of their study and provide suggestions for future research. For example, they could discuss the potential impact of variations in lighting conditions or plant growth stages on the performance of the optimized YOLOv5s model.

  6. Finally, the authors should proofread the paper carefully to ensure that it is free of grammatical and spelling errors. They should also ensure that all figures and tables are clearly labeled and easy to read.Overall, the paper presents an interesting approach for improving intelligent detection of Daylily using the YOLOv5s model with optimized depth and width parameters and the use of lightweight networks to optimize the CSPDarknet backbone network. The experimental results show significant improvements in Mean Average Precision (mAP) compared to other models and demonstrate the potential of the optimized YOLOv5s model for accurate and fast detection of Daylily in complex field environments.

    However, there are several major revision comments to consider:

    1. The paper would benefit from a clearer motivation for the research. Why is it important to achieve accurate and fast detection of Daylily in complex field environments? What are the potential applications of this research?

    2. The introduction section needs to provide more background information on the current state of research in intelligent detection of Daylily and the challenges posed by complex field environments. The authors should also discuss the limitations of existing models and why the YOLOv5s model was chosen for this study.

    3. The methodology section should be more detailed and provide step-by-step instructions for replicating the experiments. The authors should also explain why they chose to optimize the depth and width parameters of the YOLOv5s model and how they selected the Ghost, Transformer, and MobileNetv3 lightweight networks to optimize the CSPDarknet backbone network.

    4. The results section needs to be reorganized and presented in a clearer format. The authors should provide a table summarizing the performance of each model tested (YOLOv5s, YOLOv4, SSD, and Faster R-CNN) and the improvements in mAP achieved by the optimized YOLOv5s model. The authors should also provide a table summarizing the performance of the optimized YOLOv5s model with and without the use of the Transformer backbone network.

    5. The authors should discuss the limitations of their study and provide suggestions for future research. For example, they could discuss the potential impact of variations in lighting conditions or plant growth stages on the performance of the optimized YOLOv5s model.

    6. To broaden the scope of this paper, the authors should refer to some work such as:Smart contract vulnerability detection combined with multi-objective detection; A novel smart contract vulnerability detection method based on information graph and ensemble learning; SPCBIG-EC: a robust serial hybrid model for smart contract vulnerability detection
    7. Finally, the authors should proofread the paper carefully to ensure that it is free of grammatical and spelling errors. They should also ensure that all figures and tables are clearly labeled and easy to read.

Round 2

Reviewer 2 Report

Accepted.